# Determinants for male involvement in family planning and contraception in Nakawa Division, Kampala, Uganda; An urban slum qualitative study

Sarah Namee Wambete[1,2], Dorcas Serwaa[3], Edem Kojo Dzantor[4,5]*, Ararso Baru[6,7], Evelyn Poku-Agyemang[8], Margaret Wekem Kukeba[9], Yussif Bashiru[8], Oladapo O. Olayemi[10]

1 Faculty of Public Health, Department of Public Health, Nursing and Midwifery, Save The Mothers East Africa, Uganda Christian University, Mukono, Uganda, 2 Slum and Rural Health Initiative Network/Uganda, Kampala, Uganda, 3 Department of Obstetrics, Gynaecology and Newborn Health, University of Melbourne, Melbourne, Australia, 4 Department of Epidemiology and Biostatistics, Fred N. Binka School of Public Health, University of Health and Allied Sciences, Hohoe, Ghana, 5 Research and Innovation Unit, College of Nursing and Midwifery, Nalerigu, North-East Region, Ghana, 6 College of Medicine and Health Sciences, Arbaminch University, Arbaminch, Ethiopia, 7 Slum and Rural Health Initiative Network/Ethiopia, Addis Ababa, Ethiopia, 8 Department of Nursing and Midwifery, Methodist Health Training Institute, Afosu, Eastern Region, Ghana, 9 Department of Nursing, School of Nursing and Midwifery, CKT-University of Technology and Applied Sciences, Navrongo, Upper-East Region, Ghana, 10 Faculty of Clinical Medicine, Department of Obstetrics and Gynecology, College of Medicine, University of Ibadan, Ibadan, Nigeria

* edzantor21pg@sph.uhas.edu.gh

**Data Availability Statement:** Authors have indicated how the data could be accessed; Through

## Abstract

Current evidence shows that male involvement in family planning (FP) is crucial to women's contraceptive use decisions. This study explored the reasons for male involvement in FP and contraception in slum areas in Nakawa Division, Kampala, Uganda. A qualitative study was conducted among sexually active males in a slum area in Nakawa Division, Kampala. A purposive sampling technique was utilised to recruit 40 men for a Focus Group Discussion (FGDs), and 2 key informants (KI) for critical perspective interviews. A semi-structured FGDs and interview guides were used to collect the data. The FGDs were conducted in both English and the local language, Luganda. All interviews were recorded and transcribed verbatim. Transcripts for both FGDs and KI interviews were imported into the NVivo Qualitative Data Analysis Software version 10 application, and thematic data analysis was conducted. The findings show that males' involvement in FP and its decisions were minimal. The findings also show that several factors emerged as contributing to male's participation in FP and utilisation of contraceptives. Inadequate understanding of FP and contraceptives, lack of clarity on males' role in FP, unfriendly healthcare environment and community members' perceptions of male involvement in FP were reported as reasons contributing to male participation in FP and contraction. There is limited involvement of males in FP. There is a need for renewed efforts that will positively alter the factors that impact male involvement favourably. Promotion and education about FP for males will significantly address issues of limited understanding and clarity of males' role in FP services.

the ethics committee of 1. University of Ibadan/ University College Hospital (cfalade@camui.edu. ng) and 2. Ugandan National Council for Science and Technology (info@unest.go.ug). 3. Upon request made through the corresponding author. Such a request will be made available or forwarded to the Ethics Committee of the named institutions above for the necessary action to be taken. The Ethics Committee of the Institutions named do not permit the publication of audiotape and transcripts from in-depth interviews, key informants or focused group discussions as they may have the potential to result in loss of confidentiality, and privacy even in de-identified transcripts.

**Funding:** Author (SNW) received a small grant for this study from the Pan African University Life and Earth Science Institute, including Health and Agriculture (PAULESI), which covered logistic expenses such as duplication of the study tools, data collection and transportation. The funders had no role in study design, data collection and analysis, decision to publish, or preparation of the manuscript.

**Competing interests:** The authors have declared that no competing interests exist.

## Background

Males' involvement in family planning (FP) improves uptake and continuity of FP services by women [1]. Males' involvement in FP is used to describe men who participate in using FP methods and or have discussions, support, and approve the utilization of FP methods by their wives [2,3]. Evaluation of earlier studies shows that males' involvement in FP led to increased contraception uptake and important factor in improving maternal and child health outcomes [4,5]. Some studies have revealed that the relationship and communication between couples are significantly associated with better decision-making concerning contraceptive use, family size, birth spacing practices, breastfeeding, and prenatal care, all of which are critical for maternal and child health [4,5]. However, several FP programs are primarily targeted at women only [1,6].

In most developing countries, especially in the African communities, the decision-making process of a family is largely influenced by the males [1,7,8]. Evidence exist that males may be interested in FP, however, their understanding of FP is limited, making it challenging for them to discuss FP with their wives effectively [9–11]. Getting males to understand and participate in FP services and programs may significantly remove spousal barriers faced by women in the use of FP methods. FP and contraception are often regarded as women's business by males and addressing this perception is important to achieve the Sustainable Development Goals, particularly Goal 3 (Good Health and Wellbeing) [12,13].

In Uganda, the total fertility rate (TFR) stands at 5.4 children per woman, with women in rural areas more likely to have more children than their counterparts in urban areas [14,15]. The contraceptive use prevalence rate (CPR) is 39%, and the use of modern contraception is higher among women in urban areas (41%) than among women in rural areas (33%) [14,15]. Although the statistics demonstrate that urban dwellers have better access to modern contraceptives than rural dwellers, this may not be entirely true because of the varied living conditions that exist between formal and slum settlements in the urban areas. About 60% of Uganda's population is slum dwellers with the Nkwakwa Division having the largest proportion of urban populations in slum conditions [16,17]. As typical of a slum settlement, about 16 and 27% of the population of the Nkwakwa Division live 5 km or more to the nearest public health facility [16,17]. Significant disparities in FP and contraceptive use among the urban rich (48.5%) and the urban poor (21.9%) exist in Uganda [18]. Generally, the unmet need for FP exceeds 33% [19], which is higher than the national target of 10% [20] and the demand for FP services among poor households ranged between 56% in 2006 and 65% in 2016 [21].

The role of males as heads and key decision makers of the family in the African context [1,7,8] can be leveraged and maximized to address the high TFR and low contraceptive use in Uganda and slum areas. However, males' involvement in FP services remains low (40%) in Uganda [22] and the current understanding of male involvement in FP and contraceptive use among slum dwellers has not been fully explored. The understanding of males' interest in FP and their influence on women's use of available FP methods and contraception has broader implications for service uptake and changes in service delivery. To address this gap, we assessed the factors that influence males to get involved in FP in Nakawa, Uganda. The Ugandan Ministry of Health and other agencies in FP and contraception will benefit from the findings of this study in the design of FP interventions that target and support urban slum settings.

## Aim

To explore reasons for male involvement in FP in Nakawa Division, Kampala, Uganda, using a qualitative study technique.

## Methods and materials

### Study setting

The study was conducted in the Nakawa division, the largest of the city's five administrative divisions in the Kampala district. The division is located in the eastern part of Kampala city with coordinates 0°20'00.0"N, 32°37'00.0"E. It has 23 parishes and 648 villages. Nakawa is the most recognised division of Kampala because of its strategic location (a highway between Kampala and Jinja) but primarily because of Nakawa Market, the second biggest in the country. Nakawa Division is also the largest division in Kampala. The 2019 national census estimated the division's population at 317,023, with 163,594 females and 153,429 males. The population growth rate in 2019 was 4.8 per cent, the total fertility rate was 5.1 per cent, and the maternal mortality rate was 116 per 100,000 live births [23].

### Study design, population and sampling

A qualitative design was applied in recruiting males between 18 and 45 from four of the fourteen neighbourhood areas in Nakawa Division: Ntinda, Mutungo, Naguru and Kyambogo between March and June 2019. Purposive sampling was used to select ten (10) participants from each neighbourhood based on the eligibility criteria. The eligibility criteria included being a male within the age stipulated, being sexually active and consenting to participate in the study. To screen for sexually active individuals, participants were assessed on their sexual history and activities, number of sexual partners in the last 6–12 months, history of getting a woman pregnant, and how consistent or frequently has condoms been used in the previous 6 to 12 months. The exclusion criteria consisted of unwillingness to take part in the survey. We ensured the diversity of the participants by including participants within different age groups, diverse professions, educational backgrounds, and socioeconomic and sociocultural backgrounds.

Convenient sampling was used to select two key informants (health professionals to provide expert opinions on male involvement in family planning. One of the key informants was a Kiswa Kampala Capital City Authority Health Centre III health worker who had good knowledge about contraceptives and provided FP services and counselling for at least ten (10) years. The other expert was the Local Village Council (LC) chairman, the political leaders and heads of small communities within the districts of Kampala, who also make decisions in their areas of jurisdiction.

### Participants recruitment

The Local Council (LC) chairpersons were contacted two months before the data collection to mobilise eligible participants in their respective neighbourhoods for the FGDs. One researcher and two research assistants visited the selected study sites within two weeks to provide the study information such as purpose, risk, and significance to participants and who to report their decision to either or not participate in the study. The LC Chairmen required eligible participants to refer other participants to the survey. This snowball sampling helped find participants whose experiences were relevant to the study. The choice of a new subject was guided by the aim and objectives of the study and the eligibility criteria. A conference call was organised for those who consented to participate to help identify a mutually agreed location, date and time suitable for the FGD.

### Data collection

The data required to attain the objective of this study was mainly about the reasons for male involvement in FP and contraceptive usage. Although there are several techniques for

generating this data, focus group discussion and key informant interviews were employed for two main reasons: participants would be comfortable discussing their perspectives as there are other males with them, which prevents power imbalance between the researchers and the participants due to researchers' knowledge, and also help participants remember and reflect on their perspectives.

A semi-structured focus group discussion (FGD) guide designed by the researchers based on the existing literature was used to explore male involvement in FP. The FGD guide was formulated in English and translated into the local language (Luganda) by a professional translator using forward and back translations. The FGD guide covered six (6) primary themes, including (1) sociodemographic characteristics, (2) knowledge of FP and contraceptives, (3) utilisation of contraceptives and side effects, (4) perception of males' roles in FP, (5) health facility influences of FP and contraception and (6) community influence of males' participation in FP and contraception.

The FGDs were conducted in person by the researchers in community-based venues that the participants preferred. Each FGD consisted of ten participants seated in a semi-circle with the facilitator and note-taker (s) in the middle of the participants. The completion of demographic information preceded all the interviews. With participants' permission, conversations were audio-recorded, and field notes were taken. Each FGD began with the facilitator discussing the purpose of the study and setting ground rules such as respecting each other's views, allowing only one person to speak at a time, and not naming people or organisations. The discussions were held in either Luganda or English based on the participant's language preferences. Additionally, study participants were encouraged to answer the questions freely and share their opinions and experiences during the discussions. Four (4) FGDs were conducted, each FGD lasting between 45 to 60 minutes. The FDGs facilitated the attainment of in-depth data, capturing of group dynamics and interactions that were essential address the research objectives. A critical informant interview guide was used to collect data from the key informants. The Key Informant Interviews (KIIs) were also conducted and recorded in a favourable environment (a room without noise). The interviews were in either Luganda or English based on the participant's language preferences. The interviews were conducted by the principal investigator (the first author). Each KII began with the facilitator discussing the purpose of the study. Consent was sought for using an audio recorder with the assurance that the recordings would be kept confidential. Each KII lasted 45 to 60 minutes.

Saturation was met when no new information or additional information, themes, or insights emerged from the responses given by participants during the interviews. As recommended in a systematic review, between 4–8 FGDs are required to achieve data saturation [24]. Therefore, in this study, we used 4 FGDs with 10 participants each to reach data saturation. Again, each participant (that is, 40 individuals) was made to contribute to the discussions to gather sufficient data to address the research questions and objectives adequately. Further data collection was unlikely to yield substantially different findings or deepen the understanding of the phenomenon under study.

## Data analysis

Data processing and analysis were done iteratively with data collection, which helped to identify emerging themes and subthemes that directed subsequent interviews and aided in identifying data saturation. After carefully listening to each interview by the first author, those conducted in English were transcribed, whilst those in the local dialect were translated to English before transcription. Although the first author is bilingual, linguistic experts verified the translated transcripts to ensure accuracy. NVivo Qualitative Data Analysis Software

version 12 was used to manage the data. All transcripts were coded, read, and re-read to ensure all the salient points were captured. Codes were derived from the data rather than using a previously conceived coding framework to avoid truncating emerging ideas. Two persons conducted the inter-coder reliability to independently measure the consistency or agreement by assessing the same data and content. This was done to achieve high inter-coder reliability and to arrive at similar conclusions or categorisations from the data. Then, during discussions among all the authors, the coding framework was reviewed and discussed as applied to the dataset. Thematic analysis was conducted. The theme development was done by identifying and defining themes from the transcribed interviews through an inductive and deductive approach. Pseudonyms were used in the excerpts to protect the identity of the participants, and quotes from participants were used to support subthemes in reporting the study's findings.

**Ethical consideration.** The study's purpose, general content and nature were explained in a language suitable to each respondent. The respondents were informed about the right to be involved or refuse to participate in the study, the right to withdraw at any time during the FDG without needing to disclose the reason for dropping out and were assured that the data would be handled exclusively by the investigators. There was no relationship between the researchers and the participants, and only SNW had access to identifiable information. Written informed consent of individual participants was sought; participants either signed or thumb-printed to give their consent before the study's commencement and were assured of the confidentiality of their data.

This study was conducted based on the Helsinki Declaration. The study protocol, consent forms and participant information material were reviewed and approved by the University of Ibadan/University College Hospital Ethics Committee (UI/EC/18/0635). The Ugandan National Council for Science and Technology approved the research and assigned a reference number (UNCST: SS 4921). Permission to conduct the study in Nakawa Division was obtained from Kampala Capital City Authority, the District Health Officer, the District Health Inspector, the Chief Administrative Officer and the Local council chairpersons from each specified neighbourhood area where the study was conducted.

## Results

### Sociodemographic characteristics

A total of 40 male participants were recruited for the interviews. Most of the participants were in the age groups of 26–30 (27.5%), married (75.0%), and practised monogamous marriage (85.0%). Forty-five per cent (45.0%) of the participants and their female partners (27.5%) had secondary education. About 75.5% of the participants were Christians and urban dwellers (47.5%) (**Table 1**).

**Themes and subthemes.** Five major themes emerged from the data included males' knowledge of FP and contraceptives, utilisation of contraceptives, perception of males' roles in FP, health facility-related factors and the community perceptions of males' participation in FP (Table 2) demonstrating factors that impact males' involvement in FP and contraceptives use.

### Knowledge of family planning and contraceptives

The study began with exploring the participants' Knowledge of FP. Most participants in the study had some knowledge about FP and contraceptive methods, as shown by the statements of FGD 1 and 4.

**Table 1. Sociodemographic characteristics.**

| Variable | Parameters | Frequency | % |
|---|---|---|---|
| Age | 18–25 | 2 | 5.0 |
| | 26–30 | 11 | 27.5 |
| | 31–35 | 7 | 17.5 |
| | 36–40 | 9 | 22.5 |
| | 41–45 | 7 | 17.5 |
| | 46–50 | 4 | 10.0 |
| Marital status | Married | 30 | 75.0 |
| | Not married | 7 | 17.5 |
| | Separated | 3 | 7.50 |
| Type of marriage | Polygamous | 6 | 15.0 |
| | Monogamous | 34 | 85.0 |
| Level of education | No formal education | 3 | 7.5 |
| | Primary | 7 | 17.5 |
| | Secondary | 18 | 45.0 |
| | Tertiary | 12 | 30.0 |
| Level of female partner's education | No formal education | 9 | 22.5 |
| | Primary | 4 | 10.0 |
| | Secondary | 11 | 27.5 |
| | Tertiary | 9 | 22.5 |
| | I do not know | 7 | 17.5 |
| Religion | Christian | 23 | 57.5 |
| | Muslim | 17 | 42.5 |
| Area of residence | Urban | 19 | 47.5 |
| | Semi-urban | 13 | 32.5 |
| | Rural | 8 | 20.0 |

"*So, what I know about family planning is that it helps so that the children are spaced for parents to cater for them properly*" (35 years FGD 1 participant)

"*It is something used for protection with regards to birth so that an individual can*

**Table 2. Themes and subthemes generated from interviews.**

| Themes | Subthemes |
|---|---|
| Males' knowledge of family planning and contraceptives | • Understanding of family planning<br>• Knowledge of and dealing with side effects of contraceptives<br>• Males' poor attitude towards family planning and contraceptive use.<br>• Contraceptive use promotes promiscuity and extramarital affairs |
| Utilisation of contraceptives | • Side effects<br>• Poverty |
| Perception of males' roles in family planning, | • Poor attitude toward male involvement in Family Planning<br>• Limited knowledge of the roles of men in Family Planning |
| Health facility-related factors | • Family planning clinic environment<br>• The behaviours of health workers and service delivery |
| Community factors for males' participation in family planning | • Community's perception of males' participation in family planning<br>• Sociocultural and religious factors for men's involvement in family planning |

*space his children and give birth to a number that he can easily cater for"* *(40 years FGD2 participant)*

Participants mentioned some side effects of contraceptive use, such as weight gain, mood changes, breast tenderness, and changes in menstrual patterns relative to the method of contraception.

"*I understand some women experience nausea, weight gain, mood changes, breast tenderness, and changes in menstrual patterns when they take contraceptives, but the side effects can also depend on the type of contraceptive method that the person takes.*" (Age 35, FGD 3 participant)

However, some of the participants' knowledge was inaccurate and limited. For instance, whilst participants such as FGD4 claimed that FP is entirely a woman's concern, some men had no understanding of how to deal with the side effects of contraceptives.

"*Family planning is something a woman should take care of. I have also heard that women give birth to lame (disabled) babies after discontinuing the use of family planning to get pregnant. Some may also delay conceiving another child after discontinuation of the method. Some methods like the IUD cause cervical cancer, and condoms tend to widen the vagina more, and sex is normally fun with a tighter vagina"* (Age 35, FGD4 participant)

FGD 3 depicts their understanding of contraceptive side effects and ways to deal with them in the excerpts below.

"*It has side effects, especially on women in a sense that my girlfriend went in for the one they insert on the arm, and she used to bleed every two weeks, and she used to complain of her heart, so we went to remove it and tried the injection, and that was good for her. Abstaining was difficult when she was bleeding; I used to have another person. I suggest giving your partner time but moving out to look for extramarital sex. However, you cannot show her you are moving out with other women."* (Age 19, FGD 3 participant)

Although the men appear to have some knowledge of contraceptive use, there also exist some misconceptions and myths surrounding contraceptive use. Participants reported beliefs such as contraceptive use promoting promiscuity and extramarital affairs in women, contributing to low male involvement in family planning. Consequently, some of the men said that they were inclined to encourage and ensure that their mistresses (girlfriends) used contraceptives but would not allow their wives to use them. Most respondents with these misconceptions were young men aged 20–30 years and less educated. One of the FGD participants remarked that;

"*Contraceptives are mainly for men having extramarital affairs because they only want to have children with the main wife; therefore, for the side chicks, they must prevent pregnancy. Sometimes, women also lie and hold us responsible for pregnancies that we are not responsible for to get a man involved. Some women have not yet decided on the man they want to be with permanently and are still sleeping with many men. They play us, lie to us, embarrass and disrespect us. Contraceptive use promotes promiscuity among women. If she is in family planning and knows she cannot get pregnant, she will have many sexual partners to fulfil different*

*purposes. In addition, women lose respect for their husbands when on family planning, knowing they can get pregnant when they want."* (Age 22, FGD1 participant).

Participant FGD1 also suggests that men's negative attitude towards contraceptives stems from the fact that contraceptives give women the power to control when to be pregnant, thereby making them disrespect their husbands. The findings show that though there is some knowledge, this is limited and clouded with misconceptions.

**Utilisation of contraceptives.** Men's perspectives on the utilisation of contraceptives also emerged as a theme. Surprisingly, most of the participants did not use or support the use of contraceptives. The high underutilisation of contraceptives was reportedly associated with side effects related to their use. As emphasised by FGD 2 participants, abnormal vaginal bleeding, reduced libido, odours and discharge as reasons for not promoting contraceptive use.

*"Some women bleed for months while on family planning. Others gain so much weight and look unattractive; some may not see their periods for a whole year. Those condoms provided by government hospitals make (. . .) becomes unbearable. Another thing about family planning is that women no longer want to have (. . .) as frequently as possible . . . . . . ."* (Age 32, FGD 2 participant)

This FGD 2 participant also seems to suggest that contraceptives such as condoms impact sexual experience negatively as they allege that it causes foul odour in the vagina. Indeed, it is clear that perceptions like this could affect the promotion and use of contraceptives among men. Key informant 2 (KI1) also added that the reluctance to promote contraceptives among some health and family planning providers lies in the inability of some users to follow the instructions of the providers.

*"Young women often use these methods but do not follow instructions well; therefore, they end up getting pregnant from poor use of the contraceptive and then proceed to have unsafe abortions."* (Key Informant 2)

The cost of contraceptives also appears to be a reason why some men do not promote the use of contraceptives. As stated by the FGD 2, some of the contraceptives appear to be out of reach of the finances of some participants, as they indicate that the price keeps increasing, making them inaccessible.

*"The prices of these contraceptive methods keep increasing, and they are inaccessible to us. See, I am a crop farmer; where am I supposed to get money to support the children and constantly support my wife by using modern contraceptive methods?" [Age 40, FGD 2 participant]*

For the few participants who supported contraceptive use, some purported that their women had an entrenched desire to use modern methods. However, the men were unwilling to use the artificial methods (these may include hormonal methods (oral contraceptive pills and Depo-Provera), the intrauterine device, barrier methods (the latex condom and the diaphragm), and sterilization (tubal ligation and vasectomy) [25] and preferred the natural method, as shown by FGD 4.

*After giving birth to our second child, she asked if she could start using the injections not to get pregnant again soon. I was against it because of the side effects I used to hear about from my friends. But she insisted, and I finally let her do as she wanted* [Age 37, FGD 4]

**Perceptions towards males' involvement in family planning.** Although male involvement in family planning has improved, men's involvement is inadequate to yield the potential outcomes of men's involvement in family planning. The participants in this study disclosed that contraception-related decisions have always been the sole responsibility of women, and it was only recently that men's involvement is being promoted. However, some participants assert that the provision of resources to support women to engage in family planning practices has been men's responsibility, as one participant remarked:

"*Men in the past were never really involved with family planning. It was only up to the women to take care of it themselves. It has slightly changed because we are more involved, but several are still uninvolved. No men we have seen go to the family planning clinic with their partners. However, for most of us, if we have made a decision with our partners and provided the financial support and transport, we do not need to go to the family planning clinic with our partners." (Age 24, FGD3 Participant)*

". . . . .*so, we only provide financial support for the method and transport to the family planning clinic. We are also swamped working and looking for money; we do not have time to go to the Family planning clinic"* (Age 42, FGD 4 Participant)

Although participants reported providing financial support for their partners to access FP services, many indicated that they never considered accompanying their partners (wives) to the health centres to access their services. Some participants, however, acknowledge that financial support alone is enough to support their partners in practising FP as remarked by some of the participants:

"*I have not seen men in this community accompany their wives to the family planning clinic, except for one man who was a family friend of mine. He was very eager to know more about family planning, so he would always go with his wife to the family planning clinic." (KI2)*

It was noted that most FGD participants showed no interest in actively participating in their partners' family planning decisions, as insinuated by participants FGD and critical informant K12. Indeed, some of the participants were very upfront with their lack of support for the use of contraceptives, as expressed by a participant in FGD 1.

"*Male involvement in Family planning is still a new concept; it is mainly a women's issue. I don't think we should permit women to initiate family planning because of the side effects. Loss of libido is one of the toughest side effects for some of us. Imagine your wife having sex with you just for the sake of it, and she doesn't even get satisfied." (Age 27, FGD1 participant).*

A key informant indicated that women whose partners (males) supported their decisions and accompanied them to the FP clinic are happier than their counterparts. The implication of men's reluctance to support their wives in FP use has, in some instances, resorted to the covert use of contraceptives by many women mainly because their partners are not supportive. A key informant remarked this:

"*Women are normally pleased to have their husbands participate. I see the ones who come with their husbands to the family planning clinic, and it gives me so much joy because we recommend male partner support at this health centre. The women whose partners do not participate are often unhappy when they narrate their ordeals. Some women resort to hiding*

*contraceptive use because their partners disapprove, and that saddens me too."* (Key Informant 1).

**Health facility-related factors.** One of the themes from the data demonstrated that health facilities contributed to men's level of involvement in FP. Most participants reported that the FP clinic environments were unfriendly to men and did not acknowledge them as FP clients. A participant asserted that male-only information and gender-focused discussions were absent at the clinic, making it uninteresting for them to participate.

*"Men are not well received at the Family planning clinics. We are not even acknowledged as users of contraceptives. All health education efforts are directed towards women; the family planning clinics are very women-friendly and sometimes younger men, especially those below 21 years of age, are denied access to information and services"* (Age 18, FGD 1 participant)

Contrary to the plights of the FGD participants, a key informant stated that the FP clinics are very welcoming to men and have been carrying out activities that involve men as agents of change and are aimed at encouraging male partner involvement. She revealed that:

*"At our health centre, we encourage male involvement, especially in the family planning counselling sessions. We even give couples priority when providing services. Those women who come with their husbands will be attended to promptly"* (Key Informant 1)

Several FGD participants also reported that the health workers were unfriendly towards men and lacked the requisite skills and knowledge to provide FP and counselling services. Some participants believed that their presence with the partners caused the nurses to inflate the cost of the methods. Also, some providers hide relevant information on the side effects of those methods, and others do not check for compatibility. Again, the cost of contraceptive methods affected service delivery. Below is the narration of the FGD 2 participant:

*"The health workers do not have enough knowledge and education to provide family planning services. They can recommend a method and decide on it without knowing what methods are compatible with women's bodies. They also tend to recommend costly methods for our wives, yet we cannot afford them. When they see women coming with their partners, they think they have more money. As if that is not enough, they do not counsel the women well enough on the right method to choose and do not tell them that certain medical conditions are not compatible with some family planning methods. They also do not tell the women everything about the method and the expected side effects, and they catch the women by surprise. They are also rude mostly because the government does not pay them and have several problems they project onto us."* (Age 38, FGD 2 participant).

**Community factors for male participation in family planning.** The FGD participants revealed that community perceptions and cultural aspects prevented them from being involved in FP. For participants who actively engaged in FP, they reported that the community (or society) discourage its usage. In some neighbourhood areas, it was utterly unacceptable to accompany their wives to the family planning clinics. However, they were smart enough not to pay attention to community perceptions and cultural beliefs. Two participants from FGD 2 and 3 narrated:

"*The culture is negative about family planning, but religions teach us more about family planning and encourage it as a strategy to save money and manage only the children we can support. I find it very stupid to have nine or ten children, and none of them is in school because I cannot afford them. That would burden the children because they would become useless and have a difficult life if they didn't attend school. Therefore, it is not wise to follow a culture that doesn't encourage family planning.*" (Age 35, FGD 2 participant)

"*The culture does not deem going to the family planning clinic with our women acceptable, but many of us are forced to go. However, many still do not go because of cultural value*s and *low awareness. In addition, the culture maintains that white people are fighting the African culture and want to decrease our population, hence promoting family planning.*" (Age 23, FGD3 Participant)

A key informant also revealed that some religions (faiths) are entirely against family planning and, therefore, prevent some men from being adequately involved; however, the critical key informant asserted that religion is not always right about some of the beliefs and practices. He stated:

"*As a Muslim, my religion prohibits family planning, but we consult religious leaders who give us solutions on how to plan for our children and space them well without using modern contraception.*" (Key Informant 2).

"*Me, I believe that God expects us to give birth as many as we can, so using family planning is like going against what God want us to be doing*". (Age 36, FGD 2 Participant)

## Discussion

The study explored the determinants of males' involvement in FP and contraception and found that FP and contraceptive knowledge, socioeconomic factors, perception of men's role, health facility and health professional-related factors impact men's involvement in FP. Knowledge is an essential factor in individuals' decisions on practices. It is therefore vital that males who are major stakeholders in FP and contraceptive use [2,26,27] have significant knowledge of FP services and contraception. Indeed, in this study, males' knowledge of FP and contraceptives was significant. This finding is important due to the study setting, which is predominately a slum area. Promoters of FP and contraceptive use can capitalize on the findings to design interventions to increase male involvement in FP services in slum areas. Generally, literature demonstrates that knowledge of FP and contraception among males is universal [6,28–31]. However, some previous studies have reported poor knowledge of FP and contraception. For example, Kaida et al. [9] and Sileo et al. [32] reported that males have limited knowledge of FP and contraceptives in Mpigi and Butambala Districts, Uganda. Similarly, Mustafa et al. [33] reported that males have limited knowledge of FP and contraceptives in Sindh, Punjab and Khyber Pakhtunkhwa provinces of Pakistan. This underscores the importance of continuous education and promotion of FP services among males to address the significant amount of misinformation that appears to impact men's involvement in FP. Levels of male's knowledge of FP and its impact on their involvement have been widely reported in the existing literature [2,34,35]. The findings suggest that FP practitioners must devise strategies to provide males with adequate quality information that equip them enough to stimulate their interest in FP, as minimal misinformation can derail a moderate positive impact, which appears to be the case in this study.

The perception of FP as a female activity makes it critical for health professionals to act in ways that motivate males and dissuade their erroneous perceptions with negative implications for involvement in FP, thus encouraging them to participate in FP activities. However, as shown in our findings, the healthcare environment can make it impossible for males to access the correct information on FP. It may discourage them from getting involved in its use. Therefore, health professionals' attitudes towards males on FP issues must be well investigated. Areas of gaps in practices that promote male involvement relative to health professionals are harnessed to facilitate male involvement in FP to the extent that the benefits of male involvement in FP can be derived to the benefit of maternal and child health. Besides, health workers are unfriendly, discouraging males from involvement; other studies suggest that some health workers lack the requisite skills and knowledge to provide FP methods and counselling services [36–38]. Therefore, assessing health professionals' knowledge and skills and providing them with such skills and strategies for getting males involved could significantly contribute to male involvement in FP services. Due to the perception of FP as a female activity, there are limited male-focused discussions on FP, creating a gap in how they could be involved, thereby limiting their involvement. Whilst it may be essential to have interactive information sessions for males and females together on FP, it may be helpful to identify males who may want personal privacy in discussing FP issues. Thus, gender-focused FP service may be essential. Males could be advocates for FP if they are strategically and adequately engaged in FP services [26,27]. There is a need for couple-based approach [32] to FP service provision, including creating male-friendly FP clinics in Uganda and globally, cannot be overemphasised as other studies, Dral et al. [39] and Kassa et al. [2] reported findings consistent with our study.

Adopting the all-inclusive approach to FP services could also be essential in addressing real and perceived issues, such as the side effects of contraceptives and misconceptions males may hold. In our study, participants indicated some effects of contraceptives, including bleeding, dizziness, menstrual disorders and losing or gaining weight as reasons for not encouraging FP usage. Although the latter is true, educating both males and females on how to prevent or minimise the potential side effects of FP services and contraceptive use could contribute to male involvement and promotion of the latter. Other participants also hold misconceptions and beliefs, such as contraceptive use promoting promiscuity and extramarital relationships among women, making them resist the use of contraceptives. Side effects, misconceptions, myths and beliefs about FP have been reported by other studies [2,13,40]. Indeed, the evidence that these factors contribute to males' low promotion of FP has also been reported [10,41]. However, evidence suggest that these factors can be addressed by empowering males to promote FP services. For instance, multiple studies have shown that education positively predicts FP usage and involvement among males [2,8,42]. Therefore, targeting males for FP services early through incorporating FP into formal and informal education systems, creating male-friendly FP service centres, and using the Ugandan media space could allow us to increase male involvement in FP. Indeed, our study showed that participants with the most misconceptions, beliefs and myths that deter the promotion of FP services are young uneducated males between the ages of 20 and 30. Nonetheless, education of males on FP services, such as the exact types and methods of contraception, side effects and how to prevent or manage them, the benefits of FP and male involvement and how services are structured, could help attract male involvement. To address the misconceptions, beliefs and myths on FP and contraceptives, Ghulam et al. [33] study recommended the use of males with adequate training and knowledge of FP and contraceptives. The study emphasized the need to make available such males in the communities to inform other males about the benefits of FP and contraception [33]. This is because, males serving as FP and contraceptive promoters are more likely to positively influence other males to be involved in FP services.

Furthermore, while there is a perception that FP is a woman's business [13], some of the participants in our study acknowledged that they were responsible for making it possible for their female partners to access FP services. For example, other studies [8,43] have also reported that while some males may not directly engage in FP services, they support females financially to obtain transport to the health facilities and to purchase FP services. However, males' involvement in FP services goes beyond financial support. It involves understanding how FP services work, spousal communication on FP and accompanying partners to FP service centres [13,42–44]. However, some of our study participants were uninterested in accompanying their female partners to FP service centres. However, some service providers views seem to suggest that women whose partners supported their decisions and accompanied them to the FP clinic are happier than others who come to the service centres alone. Another study in Uganda showed that some women support the idea of their male partners accompanying them to FP service centres [32]. The study indicated that the females were of the view that couple-based FP services increases the quality of emotional care, joint decision making and tangible support offered by their male partners. From the perspectives of the males, the study emphasized that attending FP service centres is to primarily acquire insights on women's reproductive health issues, make better decisions with their wives regarding pregnancy and FP and to consent to the use of FP methods by their wives [32].

Interestingly, others indicated that the presence of their male partners at the FP clinics adds an extra cost to FP services. This assertion has been reported in another study in Uganda [32]. Generally, males have financially supported their female spouses to procure FP services [13,45]. However, some providers may leverage on the male presence to add other services and backdoor payments for the male spouse to pay and may serve as a barrier to male involvement in FP and contraception as has been reported in other maternal healthcare services [46]. These factors asserted by our participants are critical to discouraging male involvement in FP and contraception. Healthcare providers should consciously create an environment that is welcoming for male participation in FP. It is equally essential for the Ugandan health authorities to provide in-service training on good clinical practice, such as client care and tailored FP services that will involve couples. Multiple studies have reported on health facility-related factors influencing male involvement in FP. Such factors include the behaviour of health workers, long waiting times and an unconducive environment for male involvement in FP [37,38]. A study in Uganda reported that health workers are harsh towards male teenagers seeking FP consultation and services [36]. As indicated earlier, acting friendly towards the male, providing them with the requisite knowledge and enabling their understanding of the FP services is critical.

Community and individual perceptions and cultural practices influence male involvement in FP. Our participants indicated that the perceptions and some sociocultural inclinations of the community, such as social stigma, discourage them from being actively involved in FP. This is consistent with studies elsewhere highlighting sociocultural beliefs as determinants of male involvement in FP for their female partners [35,47,48]. Some participants in our study who indicated having accompanied their female partners to FP clinics have suffered social stigma from some community members. Religion has been a major factor determining individuals' uptake of FP services [49,50], and our participants equally reported this as a barrier to engaging in FP and contraceptive use. Although religion-related discourses could be very contentious, understanding how to apply deliberate behavioural communication change processes to develop individuals' interest in practice could be explored. Indeed, there could be FP and contraceptive options that may not go against an individual's religious beliefs. Exploring communities' and individuals' religious stances and juxtaposing them with FP interventions could contribute to finding common ground for male involvement in FP services. Again, an

alternative like faith-based advocacy initiatives can be employed to address the cultural and religious barriers to FP and contraceptive use [51,52]. Studies have shown that faith-based organizations and religious leaders can be engaged and serve as advocates for FP services, policy and funding [51,52]. They need to be given the adequate knowledge, training and support based on strong evidence on FP to effectively play the role of FP advocates [51,52].

Community sensitisation activities on FP are of the essence to addressing community-level challenges to FP use and male involvement. Opinion leaders in the community, including traditional and religious leaders, should actively be engaged in promoting FP. Likewise, male-domineering groups could be targeted for FP public health education and promotional purposes. Generally, undertaking longitudinal action research with young couples at the commencement of their reproductive life may be expensive but effective way of promoting male involvement and facilitating the uptake of FP services. Such research could demonstrate the processes and benefits of FP on whole family, serving as a health and developmental intervention.

## Conclusion

The determining factors for male involvement in FP include; an understanding of FP, side effects of FP and contraceptive methods, tagging FP and contraceptive methods as women's business and misconceptions and myths about FP and contraceptives and associated costs of FP and contraceptives. The perception of the role males plays in FP, such as providing financial support towards FP and not a responsibility to accompany female partners to FP clinics. The community variables for male involvement in FP included negative community perceptions of FP and religious beliefs. There is a need to develop an influential FP health promotional policy and implementation strategy to sensitise and address the barriers to male involvement in FP initiatives in Uganda.

## Implications

The need for interventions that target male partner involvement in FP and contraception to promote better contraceptive use and birth spacing and minimise the complications due to poor birth spacing has been supported by our study. Although the factors show that simple interventions are required to facilitate male involvement, implementing such interventions may not be as simple, considering that most factors appear to be value-laden. Therefore, whilst health professionals could initiate simple activities such as being friendly and providing male-centred information, further research may be a starting point to understand how underlying value-based factors could be tackled. As demonstrated, efforts to promote formal education and introduce FP ideas early may also improve male involvement in FP. However, enrolment issues may require policy changes to facilitate it. Perhaps satellite FP clinics that are easily accessible to men as clients and female partners may benefit the course of male involvement in FP.

## Limitations

The study was limited to males and did not solicit the views of females. Therefore, some side effects of contraception asserted by our participants that their female partners suffered could not be verified. Again, the study was geographically limited to the Nakawa Division, and therefore, the use of the findings must be contextualised. Similarly, the study used FGDs to conduct the interviews among male participants and may have limited the illumination of some detailed perspectives.

## Supporting information

**S1 Checklist. STROBE statement.**
(DOC)

## Acknowledgments

Our deepest gratitude goes to Pan African University, University of Ibadan, Ugandan National Council for Science and Technology, The District Health Office of Nakawa Division and Kampala Capital City Authority for facilitating this study. Finally, our special thanks go to all respondents and data collectors in this study.

## Author Contributions

**Conceptualization:** Sarah Namee Wambete.

**Data curation:** Dorcas Serwaa, Edem Kojo Dzantor, Ararso Baru, Evelyn Poku-Agyemang, Margaret Wekem Kukeba.

**Formal analysis:** Dorcas Serwaa, Edem Kojo Dzantor, Margaret Wekem Kukeba, Yussif Bashiru, Oladapo O. Olayemi.

**Funding acquisition:** Sarah Namee Wambete.

**Investigation:** Sarah Namee Wambete.

**Methodology:** Dorcas Serwaa, Margaret Wekem Kukeba, Oladapo O. Olayemi.

**Project administration:** Sarah Namee Wambete.

**Supervision:** Oladapo O. Olayemi.

**Validation:** Edem Kojo Dzantor, Ararso Baru, Margaret Wekem Kukeba.

**Writing – original draft:** Dorcas Serwaa, Edem Kojo Dzantor, Ararso Baru, Evelyn Poku-Agyemang, Yussif Bashiru.

**Writing – review & editing:** Sarah Namee Wambete, Dorcas Serwaa, Edem Kojo Dzantor, Ararso Baru, Evelyn Poku-Agyemang, Margaret Wekem Kukeba, Yussif Bashiru, Oladapo O. Olayemi.

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
