## [Decision Letter · Decision Letter 0]

8 Sep 2023

PGPH-D-23-01275

DETERMINANTS FOR MALE INVOLVEMENT IN FAMILY PLANNING IN NAKAWA DIVISION, KAMPALA, UGANDA; A QUALITATIVE STUDY

Dear Edem Kojo Dzantor,

Thank you for submitting your manuscript to PLOS Global Public Health. After careful consideration, we feel that it has merit but does not fully meet PLOS Global Public Health’s publication criteria as it currently stands. Therefore, we invite you to submit a revised version of the manuscript that addresses the points raised during the review process.

The reviewers have identified the need for major revisions, including changes to the description of methods and the findings. Further references and softening of language may also be needed in the Discussion in particular.

We look forward to receiving your revised manuscript.

Kind regards,

Michelle Lokot

Academic Editor

Journal Requirements:

Additional Editor Comments (if provided):

Reviewers' comments:

Reviewer's Responses to Questions

**Comments to the Author**

1. Does this manuscript meet PLOS Global Public Health’s publication criteria? Is the manuscript technically sound, and do the data support the conclusions? The manuscript must describe methodologically and ethically rigorous research with conclusions that are appropriately drawn based on the data presented.

Reviewer #1: Yes

Reviewer #2: Yes

2. Has the statistical analysis been performed appropriately and rigorously?

Reviewer #1: Yes

Reviewer #2: N/A

3. Have the authors made all data underlying the findings in their manuscript fully available (please refer to the Data Availability Statement at the start of the manuscript PDF file)?

Reviewer #1: Yes

Reviewer #2: No

4. Is the manuscript presented in an intelligible fashion and written in standard English?

Reviewer #1: Yes

Reviewer #2: No

5. Review Comments to the Author

Reviewer #1: This is a good topic and interesting results they can help dissuade beliefs that men are not interested. My comments are as follows.

1. The conclusion in the abstract do not line up with the findings well enough, I would suggest not adding findings here but work with what your results found.

2.Under data analysis, the sentence following the sentence with Nvivo 12 is awkward.

3. The title for table 1 is awkward. It could be read as poor, like lower socio-economic, and also is it based on an assumption that these men are not involved? Do we know they are not?

4. Table 1 needs some editing, there is inconsistency in some formatting

5. One quote is repeated almost verbatim, some women...it is on page 11 and also on page 13. Also the topic is repeated, when you report a result say everything you want to say about it, but then do not repeat it later.

6. Page 15 first para, misspelled word.

7. The topic on pg 17, women appreciating make involvement is not mentioned in the title or abstract on table of themes. I would either take it out or mention it earlier.

8. The men's general statements about not supporting family planning is also not well captured in the title and abstract and table, so its not just why they aren't interested to be involved, but the study also confirms that most do not want women using contraceptives. I would capture that better early on..Men's lack of support for FP and the reasons..

9. ALso the second statement about health facilities support to men given by a health care facility worker seems a bit off topic, I think its fine to have a lot of different topics but you need to frame it from the beginning.

10. I think health facility related factors, and the behaviors of of health workers and service delivery could be combined as one topic. And the two quotes about men being treated poorly could be put together.

11. The two quotes around money could also be put together rather than stating it once, moving to another topic, and then coming back t it..

12. The paragraph under community factors is hard to understand, is the community frowned upon? Or does the community frown upon the couple?

13. I feel like the discussion does not capture the breath of the results in the same way that the introduction and title and themes do not. Either streamline the results, or capture them...

14. I think both the discussion and the introduction could be more streamline, its good that all the distinct topics are brought up, but it circles some.

Reviewer #2: Peer review report

This qualitative study describes the determinants for male involvement in family planning in Nakawa Division, Kampala, Uganda. Authors conducted 4 FGDs with 10 participants each and two key informant interviews and reported the results. Finally, some of the main concepts of qualitative analysis such as double coding, inductive/deductive analysis, and data saturation appear to be missing. For the next round of reviews, authors should also provide line number to make it easier to offer feedback.

Authors need to make the following major and minor revisions to be considered for publication. I request the authors to proofread thoroughly and copyedit to avoid any mistakes. Several statements have been made that have no citations. I suggest even checking the apostrophe such as here in Table 1 (it should be men’s not Men ‘s) and minor typos throughout the manuscript. Please do a full spell check of the manuscript as well.

Major revisions:

BACKGROUND

• Page 5: Last paragraph is about past studies and why the current study is needed. I suggest the authors to not mix with prior evidence on why men’s involvement in FP is low. What do authors mean by not a single qualitative study from Uganda has explored this topic? Please add qualifiers such as, to our knowledge, we are not aware of any publications that have explored this topic quantitatively.

o Further, what do authors mean by “no information on change processes for male program beneficiaries”; please rephrase for clarity.

METHODS

• How did the authors screen males for sexual activity? Did you assume all men in 18-45 years were sexually active?

• Why did authors choose only 2 key informants?

• Were FGDs conducted on video conference or in-person?

• Data collection is missing several important elements:-

o What did the FGD guide cover in terms of themes and topics?

o How did authors determine 4 FGDs with 10 participants each was sufficient? Or in other words, how define how saturation was achieved.

o Please use either participants or discussants for consistency throughout.

DATA ANALYSIS

• Please cite the NVivo 12 software properly.

• How was the intercoder reliability established? How many researchers coded the data?

• Check typo, it cannot be “will be used” in this sentence “Pseudonyms have been used in the excerpts to protect the identity of the participants, and quotes from participants will be used to support subthemes in reporting the findings of the study”.

• What is the SNW acronym?

• The analysis section is missing whether the analysis approach – of coding, and whether deductive content analysis or inductive analysis was used.

RESULTS

• Although qualitative work does not require quantifying the results (for example, 10/15 perceived this…), we still need to know whether many participants, most participants, or few participants endorsed a theme or felt a particular way.

• Re: the quote, please change the words from lame babies to something else. I understand it is verbatim but perhaps the English translation should slightly to reflect the participants’ sentiments. Check whether the participant means disabled babies? “Family planning is something a woman should take care of. I have also heard that women give birth to lame babies after discontinuing the use of family planning to get pregnant.”

• Avoid using discussant, respondent, participant – and please use only of these words consistently and throughout the paper.

• On Page 12: Under the utilization of contraceptives, most of the results appears to be only from FGD 2 and some from FGD 5– it is not clear why only one FGD is being used to highlight. Please add results from other FGDs as well. Is FGD 5 a typo because my understanding is that there were only 4 FGDs in total? Does FGD number indicate the respondent within a given FGD or FGD number (as in, FGD 1, FGD 2, etc.). Please clarify and edit as needed.

• On Page 14, “were unwilling users of the artificial methods and preferred the natural method”, what do authors mean by “artificial methods”

• Please edit the title “Perceptions of the role’s men play in family planning”

• What does this mean “The participants in this student disclosed that contraception related decisions”. It looks like the paper was written in a rush without sufficient proofreading/ editing.

• It looks like the two themes “Health facility-related factors and behavior of health workers” could be combined into a single result.

• On pg 17. Under this “Community factors for men's participation in family planning”, please edit this word-- The few actively engaged in family planning reported that the community was frowned upon by society.

• On page 18, edit “A key informant also revealed that some of their religions are completely against family planning, and therefore it prevents some men from being adequately involved; however, he asserted that religion is not always correct.” Do authors mean some people of certain religious faith?

• On Page 21, “Our study underscored that women whose partners supported their decisions and accompanied them to the FP clinic are happier than others who come to the service centres alone” I don’t remember seeing any theme speaking to this result. Since your study only interviewed men, I don’t see how you can comment on women’s feelings or perceptions.

• Page 21. Similarly this sentence “Others indicated that presence of their partners at the FP clinics adds the extra cost to FP services” – this is not clear. Who are the partners? The men or the women? It’s very confusing as written.

• Page 22: “Religion has been a major factor determining individuals uptake of FP services 46,47 and this was equally reported by our participants as a barrier to engaging in FP and contraceptive use.” I did not see religion as a major deterrent to men’s engagement in FP. None of the results except one quote from a key informant that mentioned religion. Please add religious factors in the results if religion showed up as a theme.

• “Indeed, there could be FP and contraceptive options that may not ran contrary to the religious beliefs of individuals.” What does this statement mean? Edit for clarity.

Additional points:

• Socio-demographics of 40 FGD participants should be added as a table. It is currently missing.

• COREQ checklist is missing from the submission. Please submit this along with the manuscript for the next round.

Minor revisions:

• Background

o Avoid using words like alarming for CPR (Page 4).

o “Some studies have revealed that the relationship and communication between couples are significantly associated with better decision-making concerning contraceptive use, family size, birth spacing practices, breastfeeding, and prenatal care.” Please cite the literature for this statement. There are several such instances throughout the introduction.

• Also, please tone down the language “Men are the reason for most of the reproductive health issues suffered by their female partners in Uganda”. While I understand the emotion here, language for papers should avoid extreme tones.

• Remove random capitalization of words such as “Lack of Knowledge”

• Remove this—"coordinates 0°20'00.0"N, 32°37'00.0"E”

• TFR is not reported as per cent, please check and update correctly.

• Check typo or edit for clarity: Counseling for not less than years?

• “The Local Council (LC) chairpersons were conducted two months before the data collection to mobilize eligible participants in their respective neighbourhood for the FGDs” – please check this statement and edit for clarity.

6. PLOS authors have the option to publish the peer review history of their article (<a href="https://journals.plos.org/globalpublichealth/s/editorial-and-peer-review-process

---

## [Decision Letter · Decision Letter 1]

14 Feb 2024

PGPH-D-23-01275R1

DETERMINANTS FOR MALE INVOLVEMENT IN FAMILY PLANNING IN NAKAWA DIVISION, KAMPALA, UGANDA; A QUALITATIVE STUDY

Dear Dr. Dzantor,

Thank you for submitting your manuscript to PLOS Global Public Health. After careful consideration, we feel that it has merit but does not fully meet PLOS Global Public Health’s publication criteria as it currently stands. Therefore, we invite you to submit a revised version of the manuscript that addresses the points raised during the review process.

We look forward to receiving your revised manuscript.

Kind regards,

Ifunanya Clara Agu

Academic Editor

Journal Requirements:

Additional Editor Comments (if provided):

Authors should address all comments raised by the reviewer two and three.

Reviewers' comments:

Reviewer's Responses to Questions

**Comments to the Author**

1. If the authors have adequately addressed your comments raised in a previous round of review and you feel that this manuscript is now acceptable for publication, you may indicate that here to bypass the “Comments to the Author” section, enter your conflict of interest statement in the “Confidential to Editor” section, and submit your "Accept" recommendation.

Reviewer #2: All comments have been addressed

Reviewer #3: (No Response)

2. Does this manuscript meet PLOS Global Public Health’s publication criteria? Is the manuscript technically sound, and do the data support the conclusions? The manuscript must describe methodologically and ethically rigorous research with conclusions that are appropriately drawn based on the data presented.

Reviewer #2: Yes

Reviewer #3: Partly

3. Has the statistical analysis been performed appropriately and rigorously?

Reviewer #2: N/A

Reviewer #3: N/A

4. Have the authors made all data underlying the findings in their manuscript fully available (please refer to the Data Availability Statement at the start of the manuscript PDF file)?

Reviewer #2: No

Reviewer #3: No

5. Is the manuscript presented in an intelligible fashion and written in standard English?

Reviewer #2: No

Reviewer #3: Yes

6. Review Comments to the Author

Reviewer #2: In general, the manuscript has successfully incorporated feedback from the initial review. However, attention is required to address the specific minor comments outlined below. Furthermore, I recommend that the authors undertake a thorough revision of the manuscript to enhance clarity and rectify any typographical errors, ultimately elevating the overall quality of the manuscript.

Minor edits

On Page 14, “were unwilling users of the artificial methods and preferred the natural method”, what do authors mean by “artificial methods”. This should term “artificial methods” should be explained in a footnote in the manuscript because this is not a commonly used phrase in contraception/family planning. Authors replied to the comment by saying “Artificial contraceptive methods are various techniques and devices designed to prevent pregnancy by interfering with the process of conception.” Can you provide examples of such methods? I still don’t get it. Also, in addition to responding to reviewers, pl add this info in the manuscript.

On Page 23 “Others indicated that the presence of their male partners at the FP clinics adds the an extra cost to FP services”. Why does presence of male partners add an extra cost to FP services? It is still not clear.

Page 25: There is a line in the conclusion “I am tagging FP and contraceptive methods as

women's business and misconceptions and myths about FP and contraceptives.” This does not make sense. Again it seems the authors have not proofread and copyedited the manuscript sufficiently to ensure clarity in writing. Please spend sometime in proofreading and correcting small typos.

Page 25: …replace this line and add “The health-related determinants for male involvement in family planning include an unfriendly family planning clinic environment, a lack of recognition of males as users of contraceptives, insufficient gender-focused discussions, inadequate information on men’s family planning methods, and unfriendly attitudes of health workers toward males in family planning”. Currently there are several typos in the line and it is grammatically incorrect.

Reviewer #3: Overall, this is an interesting study that reveals insights into men’s participation or lack thereof in family planning in Uganda. The themes highlighted reflect notable findings and are sufficiently backed by the interview extracts. This has valuable implications to how FP interventions are delivered. The main critique here is that in the background literature, the authors emphasize the rural/urban differences in contraceptive use with rural areas having the highest number of children and least contraceptive use. Yet the study focuses on an urban population of men, and this should be included as a limitation of the study. Either the study title should adjust to reflect male participation in urban settings, or the study should include FGDs in a rural context to sufficiently explore the stipulated aim of the study. The discussion should be strengthened by expanding on other studies that agree with or are contrary to the author’s findings. Additionally, the language in some sections comes across biased and needs to be rephrased. Further comments are included below.

Background (p.4) – Do men seldom involve themselves or have family planning initiatives historically excluded men and focused on women? It is worth acknowledging this aspect as well.

Background (p.4) – “Evaluation of several early studies showed that men's involvement led to increased contraception uptake.” Include citations.

Background (p.4) – “The most frequent reason for the non-use of contraceptives is the lack of spousal support.” Include citations.

Background (p.4) - Men are the reason for most of the reproductive health issues suffered by their female partners in Uganda. This is a rather strong accusation and may lead to resistance. Rephrase. Is it the limited knowledge/culture that impacts their decision making and therefore how they influence family planning?

Aim (p.5) – Are there specific questions broadly that were used to explore male involvement? If so, include them.

Include interview questions in the appendix of the manuscript.

Study setting (p.6) – Why was Nakawa chosen as the study site? Relative to the question of low male involvement? Have SRH interventions been run in this area? Why not a rural site given the background information on women in rural areas likely having more children and less likely to use contraception than women in urban areas? Nakawa is considered urban. The rural-urban comparative would be considered critical in diversifying the participant pool.

Participant recruitment (p.7) – LC chairpersons were contacted not conducted.

Data collection (p.8) – FGD among males is a culturally sensitive topic and would have benefited from one-on-one interviews to illuminate more intimate perspectives and counter group think. Not conducting one on one interviews may need to be justified or be discussed as a potential study limitation. Include high-level examples of what types of questions were posed during the focus group discussions.

Data analysis (p.9) – it is not clear what data analysis method was employed. Is this thematic analysis? Content analysis? Clearly indicate inductive approach was taken to coding, if this is the case.

Findings (p. 11). Men’s inappropriate attitude – who judges it to be inappropriate. Rephase to men’s attitude or poor attitude towards FP and in the descriptions describe it and why it may be inappropriate. Otherwise again, this is alienating language.

Discussion p.19. It would be helpful to see literature that contradicts the study’s findings. All cited literature is consistent with the current study. Furthermore, for instance why is information on FP side effects not provided to patients based on the literature? You cite this accusation as true, why is it the case?

Discussion p.20. Take the time to identify at least what one of the citations you have included contributes to the issue of or intervention on side effects, myths or beliefs. What have some of the successful interventions looked like?

Discussion p.21. The claim around women being happy when escorted by husbands to service centers is made by one key informant’s perspective and did not come directly from the women. It is not accurate to make this claim on behalf of women unless validated by women participants.

Discussion p.22 Provide examples, beyond simply citing, of studies that show successful integration of culture/religion in FP initiatives to illuminate this point. Same applies to “an all-inclusive-approach”, what examples exist on this anywhere in the world.

7. PLOS authors have the option to publish the peer review history of their article (what does this mean?). If published, this will include your full peer review and any attached files.

**Do you want your identity to be public for this peer review?** For information about this choice, including consent withdrawal, please see our Privacy Policy.

Reviewer #2: No

Reviewer #3: No

---

## [Editor Report · Decision Letter 2]

17 Apr 2024

Determinants for Male Involvement in Family Planning and Contraception in Nakawa Division, Kampala, Uganda; An Urban Slum Qualitative Study

PGPH-D-23-01275R2

Dear Mr Dzantor,

We are pleased to inform you that your manuscript 'Determinants for Male Involvement in Family Planning and Contraception in Nakawa Division, Kampala, Uganda; An Urban Slum Qualitative Study' has been provisionally accepted for publication in PLOS Global Public Health.

Best regards,

Ifunanya Clara Agu

Academic Editor